# A Deep Learning Integrated Cairns-Blake-Dowd (CBD) Sytematic Mortality Risk Model

**Joab Odhiambo \*** , **Patrick Weke** and **Philip Ngare**

School of Mathematics, University of Nairobi, Nairobi City 30197-00100, Kenya; pweke@uonbi.ac.ke (P.W.); pngare@uonbi.ac.ke (P.N.)
\* Correspondence: joabodhiambo2022@gmail.com

**Abstract:** Many actuarial science researchers on stochastic modeling and forecasting of systematic mortality risk use Cairns-Blake-Dowd (CBD) Model (2006) due to its ability to consider the cohort effects. A three-factor stochastic mortality model has three parameters that describe the mortality trends over time when dealing with future behaviors. This study aims to predict the trends of the model, $k_t^{(2)}$ by applying the Recurrent Neural Networks within a Short-Term Long Memory (an artificial LSTM architecture) compared to traditional statistical ARIMA (p,d,q) models. The novel deep learning (machine learning) technique helps integrate the CBD model to enhance its accuracy and predictive capacity for future systematic mortality risk in countries with limited data availability, such as Kenya. The results show that Long Short-Term Memory network architecture had higher levels of precision when predicting the future systematic mortality risks than traditional methods. Ultimately, the results can be implemented by Kenyan insurance firms when modeling and forecasting systematic mortality risk helpful in the pricing of Annuities and Assurances.

**Keywords:** systematic mortality risk; deep learning; long short-term memory; CBD; recurrent neural networks

## 1. Introduction

Since the start of the 21st century, mortality rates have been decreasing steadily due to several factors such as improved medical inventions, robotic surgery, better healthcare systems, and better diets, among many other factors, see (Boo and Choi 2020; Chen 2020; Kilic 2020; Pourhomayoun and Shakibi 2020). These factors have prompted actuaries, demographers, and statisticians to think of novel ideas to do mortality modeling and forecasting for an increased level of precision in the models. While this is a good idea for the general global population, many governments, life assurance firms, and life pension companies have substantial financial losses since they cannot make precise estimations when offering financial services. Correct mortality risk estimation is vital in their financial survival, especially after the hard times of the global Covid-19 pandemic see (Pourhomayoun and Shakibi 2020) that is likely to lead to massive global economic recession affecting many nations, both first-world and third-world countries.

Today, in the actuarial literature, we have many refined techniques that many actuaries, statisticians, and demographers use when forecasting future mortality and systematic longevity risks. We estimate the complete life expectations of those who wish to buy annuities and life assurance products sold in the market. From (Lee and Carter 1992), there are many stochastic mortality models currently used when modeling and forecasting systematic mortality risk. However, these models have different strengths and weaknesses depending on the data availability and the number of available parameters that need to be determined or estimated. (Cairns et al. 2006) model improves some of the weaknesses of the (Lee and Carter 1992) model by incorporating the cohort effects and double parameter ability. Modeling a systematic mortality risk that is constant over age while preventing overlapping of the age lines during forecasting leading desirable results as a parsimonious

model was demonstrated by (Cairns et al. 2011), which leads to high certainty levels and acceptability in the results.

Many researchers, including (Hainaut 2018) in his paper, proposed a neural network capable of predicting and simulating future systematic mortality risk. During his research, the author used a neural analyzer when detecting latent time processes while directly predicting mortality. The approach did allow for identification and duplication of non-linearity observed in the changes of logit forces of mortality. In addition, (Deprez et al. 2017) used some machine learning techniques to improve the estimation process of the logit mortality risk. This work was extended by (Levantesi and Pizzorusso 2019) to the framework of mortality forecasting as in the [3] model. Furthermore, a recent paper by (Richman and Wüthrich 2018) proposed multiple-dimensional populations for (Lee and Carter 1992) model where it estimated the parameters using artificial neural networks. Many other relevant machine learning uses and applications in an actuarial field are discussed by (Castellani et al. 2018) and (Gabrielli et al. 2020), especially when looking at the future of systematic mortality risk modeling methodologies.

In this research study, we use a deep learning technique to improve the predictive capability of the (Cairns et al. 2006) model. To be more specific, our approach aims at Integrating the original (Cairns et al. 2006) formulation by the introduction of an artificial Recurrent Neural Networks with Long Short-Term Memory or LSTM architecture when forecasting future evolution of the $k_t^{(2)}$ parameter thus overcoming the challenges showed by the traditional ARIMA $(p, d, q)$ time series process. The choice of the CBD model instead of other standard mortality models is based on the fact that CBD solves the problem of cohort effect in mortality synonymous with other mortality models. In addition, it incorporates the effect of cohorts in models compared to others used in modeling of systematic mortality risk.

Using LSTM allows more coherency when determining mortality forecasts with high dynamism of observed mortality, especially when dealing with nonlinear mortality trends. To be more precise, the LTSM network is structured to help elaborate long data sequences to form a memory capable of preserving the vital relationships between the available data and every deviation within these sequences. In a similar sense, within the context of traditional time series, the LSTM gives room for predicting future mortality over time by considering the substantial influence of the historical systematic mortality risk trends before adequately reproducing it into the forecasted trend. In addition, the power of LSTM is by preserving the information over a given period, therefore blocking the older signals from slowly disappearing during processing.

While the research focuses on forecasting systematic mortality risk trends, parameter estimation methodology remains similar as for (Lee and Carter 1992). The paper does introduce a new method to mortality fitting surface as by (Hainaut 2018) that applies the use of neural networks or deep learning technique for fitting mortality rates as opposed to the conventional SVD method (Singular Value Decomposition). This study introduces a novel methodology structure based on the LSTM network when modeling future common trends of systematic mortality risk.

## 2. Cairns-Blake-Dowd (CBD) Model

**Definition 1.** *Let the (Cairns et al. 2006) be;*

$$logit\mu_{(x,t)} = \alpha_x^{(1)} k_t^{(1)} + \alpha_x^{(2)} k_t^{(2)} + \alpha_x^{(3)} w_{t-x}^{(3)} \qquad (1)$$

The cohort effect influence, $w_{t-x}^{(3)}$, for any age-specific cohort has been assumed to reduce to zero with time. $\alpha_x^{(3)}$ decreases with $x$ as opposed to being a constant i.e., $\alpha_x^{(3)} = c$ where $c$ is a constant itself. Therefore, this will give us the model as

$$logit\mu_{(x,t)} = \alpha_x^{(1)} k_t^{(1)} + \alpha_x^{(2)} k_t^{(2)} + \alpha_x^{(3)} w_{t-x}^{(3)} \qquad (2)$$

where $\alpha_x^{(1)} = 1$, $\alpha_x^{(2)} = (x - \bar{x})$, $\alpha_x^{(3)} = (x_c - x)$. With the replacement of the values, we have:

$$logit\mu_{(x,t)} = k_t^{(1)} + k_t^{(2)}(x - \bar{x}) + w_{t-x}^{(3)}(x_c - x) \qquad (3)$$

During the analysis, we have to use the constraint $\sum_{i=1}^{\infty} w_{t-x}^{(3)} = 0$ to prevent introducing the identifiability problem during the process of estimation as well as projection. The model has no problems of identification. In the original (Cairns et al. 2006) model, researchers often used $SVD$ or singular value decomposition when estimating parameters as per the 2-stage procedure. This is done by applying it to the matrix of $logit\mu_{(x,t)}$ as a way of finding values of $k_t^{(1)}$ to thus obtaining values $k_t^{(2)}$ and $(x - \bar{x})$ respectively. Secondly, to ensure that observed deaths coincide with the estimated deaths, $k_t^{(2)}$ is refitted.

**Lemma 1.** *As per the traditional (Cairns et al. 2006) formulation, $k_t^{(2)}$ is often modeled using an Auto-regressive Integrated Moving Average $(0, 1, 0)$ as;*

$$k_t^{(2)} = k_{t-1}^{(2)} + \delta + w_{t-x}^{(3)} \qquad (4)$$

*whereas $\delta$ is defined drift parameter and $w_{t-x}^{(3)}$ are the randomness term and $w_{t-x}^{(3)} \sim N(0, \sigma_k^2)$.*

## 3. The Neural Network Model

### 3.1. Artificial Neural Network Definition

**Definition 2.** *An ANN (Artificial Neural Network) is a series of algorithms that endeavors to identify underlying relationships in a given data set via a process capable of mimicking how a human brain works. Artificial neural network architecture includes neurons, the synaptic connections, which link the neurons, and learning algorithms.*

*ANN is formed through 3 categories of layers, known as hidden, input, and an output layer respectively where each one of the layers is made up of several neurons (Hassoun et al. 1995). Every unit in an artificial network obtains "proportional" information via synaptic links from many other well connected ones at the same time returning an output through using an artificial activation function that transforms these proportional totals of the input signals.*

### 3.2. Deep Learning Modeling

**Definition 3.** *Let Q denote a single neuron called perceptron defined by;*

$$Q = \Theta(Z^T y + c) \qquad (5)$$

*where $y \epsilon \mathbb{R}^{\subsetneq}$ is the input and $Z \epsilon \mathbb{R}^{\subsetneq}$ is the connected synaptic weight, $\psi \epsilon \mathbb{N}$ are numbers of the input signals and $\Theta$ is the activation function. We represent this term c as the bias that is associated with the model known as activation verge or threshold. The user must note that the function, $\Theta$, should have a differential because the learning equations have gradients (Minsky and Papert 2017).*

We introduce Multilayer Perceptron ( MLP ) used in nonlinear separable problems such as Exclusive or (XOR) since ANN with a single layer is always inappropriate, thus solving the stated problem. In addition, most neurons in MLP are predisposed on a wide variety of layers, with every unit fully connected to those of the preceding layer, as illustrated by (Goodfellow et al. 2016). The synapses connect units by defining different types of available networks in the system. In an ANN classical pattern like feed-forward ANN, the information moves in a unilateral direction from an input to an output layer at the same time the Recurrent Neural Networks (commonly known as RNNs) processes the information cyclically using the extra synapses to ensure that the reprocessed out is as a result of the entire elaboration process.

Figure 1 below shows the standard representation of feed-forward ANN. A neuron is represented in every node, connected from one to the other using arcs representing all synapses. Additionally, the graph represents the general input, latent, as well as output variables.

The Schematical view of an artificial neural network (ANN) below has circles representing neurons with lines representing synapses. The Synapses take the individual inputs before multiplying them by a "weight" commonly known as input "strength" to determine the general output. In addition, Neurons are added to these outputs from all available synapses before applying the activation function.

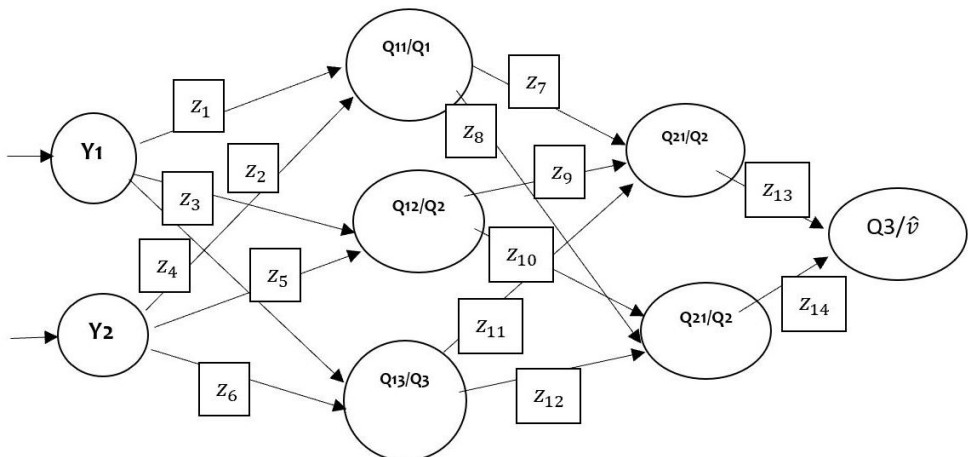

**Figure 1.** A Normal representation of feed-forward ANN.

**Definition 4.** *From the output, let $Q \epsilon \mathbb{R}^{k_h}$ denote a generic hidden layer having $k_h$ neurons defined as;*

$$Q_1 = \Theta(Z^T \boldsymbol{y} + \boldsymbol{c}) \tag{6}$$

*where $Z \epsilon R^{\psi * k_h}$ is defined as a weight matrix and $c \epsilon R^{k_h}$ is called the biases vector. According to MLP scheme, the hidden layer output becomes the input instrument for the following layer.*

**Lemma 2.** *Considering a given problem of regression, where $f \epsilon \mathbb{N}$ defined as the number hidden layers, then the output of $\hat{y} \epsilon \mathbb{R}$ can be calculated by:*

$$Q_1 = \Theta_1(Z_1^T \boldsymbol{y} + \boldsymbol{c_1})$$

$$Q_2 = \Theta_2(Z_2^T Q_1 + \boldsymbol{c_2})$$

$$Q_3 = \Theta_3(Z_3^T Q_2 + \boldsymbol{c_3})$$

$$.........$$

$$\hat{y} = \Theta_f(Z_f^T Q_{f-1} + \boldsymbol{c_f})$$

*where $Z_1, Z_2, Z_3, ...Z_f$ denote weight matrix vectors, $c_1, c_2, c_3, ...c_f$ denote bias vectors, and $\Phi_1, \Phi_2, \Phi_3, ...\Phi_f$ denote activation functions that needs not be different from one another.*

It is vital to note that all measurements of the weight matrices and bias vectors do rely on the unit number within the hidden layers; hence, by enhancing these hidden layers in numbers, the abstraction levels of the input data also increase significantly.

*3.3. Backward Propagation of Errors*

**Definition 5.** *Backpropagation is an algorithm used for supervised learning of artificial neural networks through gradient descent. Provided an artificial neural network (ANN) and an error function, this method is capable of calculating the error function gradient with respect to the respective weights of neural networks.*

*ANN training involves the use of a given unconstrained optimization problem with the aim of minimizing a function within the high dimensional space. We start by defining a loss function as:*

$$B = \frac{\sum_{i=1}^{f} (v_i - \hat{v})^2}{2} \tag{7}$$

This loss function measures the deviations of predicted values $\hat{v}$ from the observed ones $v$ i.e., it obtains the absolute error terms between these predicted values of $\hat{v}$ as well as observed values of $v$. The quantity $B$ also relies on the weights of the matrices namely $Z_1, Z_2, Z_3, ... Z_f$, which ultimately influences the values of predicted $\hat{v}$. Consequently, the aim of the method is to find the exact synaptic weight values, which minimizes the value of quantity $B$.

While machine learning has many algorithms applied in its application, backpropagation is among the most commonly used feed-forward training ANNs. The algorithm works by comparing the predicted values versus the expected ones according to modifying the synaptic weights through back-propagating the loss function's gradient.

From Figure 1, the procedure continuously alternates forward with backward propagation in the following steps, namely in the forward step, the predicted values of $\hat{v}$ are calculated by fixing the respective synaptic weights, and in this backward step, the adjust weights thus reducing the error $B$ of the network. It is important to note that ANN can iteratively perform both forward and backward propagation by modifying the weights to find the combination, which minimizes the overall loss function.

**Definition 6.** *Analytically, backpropagation algorithm updates all weights of $Z_f$ in the last layer by the rule of delta as follows;*

$$\Delta Z_f = -i \frac{\partial B}{\partial Z_f'} \tag{8}$$

*where i is called the learning rate. As for other preceding layers, we differentiate using product or chain rule of differentiation. The other weights matrix $Z_{f-1}$ are determined as:*

$$\Delta Z_{f-1} = -i \frac{\partial B}{\partial Q_{f-1}} * \frac{\partial Q_{f-1}}{\partial Z_{f-1}} \tag{9}$$

*and the process continuous on to many other layers in the system.*

We look into the same idea in a figurative way, just like a gradient or slope descent similar to a "climbing down a steep hill" so long as it reaches a local minimum or global limit. However, at every update, the search does move in the gradient's opposite direction while the slope of the gradient and learning rate is determined by the Movement amplitude (Baydin et al. 2017). Moreover, the choice of rate i is a vital element, as a small value can lead to several iterations simultaneously; larger values might permit convergence, especially to a global minimum.

We choose from a wide range of architecture, including the hidden layers numbers, units for every layer, and the hyper-parameter values like learning rate, epochs, and activation function, which remain another heuristic problem for ANN users. It is important to note that the choice will always depend on the data type available, which might be a difficult step or just easy. An initial round of the hyper-parameters tuning, especially before the testing, might be highly needed. Additional extensive descriptions of ANNs and back-propagation algorithms are explained well (Alpaydin 2016) and (James et al. 2013).

*3.4. Recurrent Neural Network Using a Long Short-Term Memory Architecture*

**Proposition 1.** *We incorporate the concept of Deep Learning techniques in stochastic mortality modelling to increase their predictability and forecasting accuracy.*

**Proof.** The feedforward ANNs, which always represent a powerful tool for analysis, can be insufficient when effectively managing time sequences of the available data. However, the recurrent connections between nodes that have featured the RNNs allow for an active analysis of the given sequential data. Nevertheless, through applying the given RNN structure, we often face the massive problem of gradients disappearing and weights change, before becoming tiny fast to show no effect. Consequently, the network will gradually lose its capability of learning from the past to become operationally insufficient for the more prolonged data sequences analysis and thus helping in making excellent predictions. It is why we say that RNNs possess a short memory only. □

As a way of overcoming the stated problem, (Hochreiter and Schmidhuber 1997) had come up with the Long Short-Term Memory, commonly abbreviated as LSTM. The LSTM is a version of RNN whose architecture can allow considerate relationships between the sequence of data, even if it happens in the long run, thus eradicating the vanishing gradient problem in the process. Similarly, RNNs need both long- and short-memory, thus managing to generate an extraordinary performance in the analysis of time series. However, several improvements in the original work, LSTM, have been improved through a series of studies such as (Bahdanau et al. 2014) and (Cho et al. 2014). Ultimately, one can define an excellently composed basic structure as vanilla LSTM.

**Definition 7.** *From Figure 2, Let $f_t = g_t$, $i_t = r_t$ and $o_t$ denote he output that would be important in RNN analysis. Let output of the auxiliary-output gate be defined as;*

$$g_t = v(Z_f y_t + U_f Q_{t-1} + c_f) \tag{10}$$

$$r_t = v(Z_i y_t + U_i Q_{t-1} + c_i) \tag{11}$$

$$o_t = v(Z_o y_t + U_o Q_{t-1} + c_o) \tag{12}$$

$$j_t = v(Z_j y_t + U_j Q_{t-1} + c_j) \tag{13}$$

The forget gate output $g_t$ as defined by Equation (10), illustrates facts from the preceding cell state as well as the one originating from the present input are mixed within a nonlinear way through a sigmoid activation function. Afterwards, $g_t$ is mixed through a point-wise product especially within its previous memory state $c(t-1)$. Its input gate $r_t$, as defined in Equation (11), uses an active sigmoid activation, which permitting for decisions when information is received before it is updated. The output gate $o_t$, as defined in Equation (12), plays the role of preventing non-significant memory content transmission that is stored information within the other blocks. Its role as a sigmoid function is to pass appropriate memory information. As a way of regulating processed data flow, the input gate it does combines with that derived from all linked auxiliary NN $j_t$ as defined in Equation (13).

**Definition 8.** *Let denote the entire input block processing procedure that participates in construction of the present memory cell state as:*

$$c(t) = c(t-1) \circledast g_t + r_t \circledast j_t$$

To get the current output, which is a combination in between the defined function in above equation;

$$Q = \Phi(c(t)) * o_t \tag{14}$$

From Equation (14), this LSTM architecture offers an outstanding tool when dealing with forecasting time series, particularly in cases of longer time lag connections, catching randomness, and management of the noise. Nevertheless, any user of LSTM, just ANNs in general, must have the face of the classical problems that concern the hyperparameters choices.

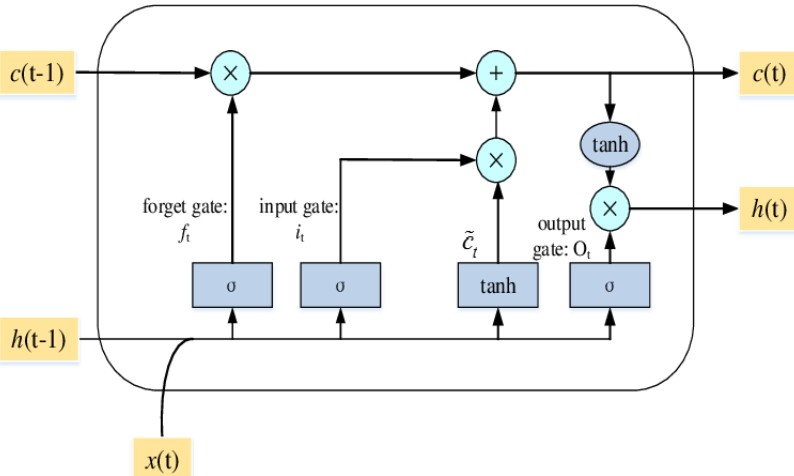

**Figure 2.** A LSTM Block Structure with Its Internal Information Forward Flow Design.

### 4. Mathematical Application and Results

In this area, we introduce the LSTM and RNN architectures within the standard scheme of the CBD model. More distinctly, the study's objective is to exploit the advantages and functionalities of the LSTM architecture to improve the CBD model predictive capacity. For this aim, we design several experiments to test LSTM skills in forecasting future systematic mortality risk over time before comparing its performance with the results derived from the model of ARIMA.

Thus, the analysis of the study will concern on the time index $k_t^{(2)}$ trend prediction, bearing in mind the ARIMA $(p, d, q)$ model as the forecasted benchmark, whereas other parameters $k_t^{(1)}$ and $(x - \bar{x})$ are determined as per the estimation method by (Cairns et al. 2006).

Distinctly, the CBD model that applies a simple random walk process with drift is vital to calibrate the best ARIMA (p,d,q), as illustrated by (Hyndman and Khandakar 2007). This procedure checks the time series stationarity in the initial round using a suitable unitary root test before choosing the differencing order d. The 2nd stage determines the autoregressive best values and moving average order, like p and q, respectively, using exact information criteria for AIC or BIC. In most cases, the implemented algorithm utilizing the function, which is present in the python package for forecasting (Hyndman and Khandakar 2007); and (Bauer et al. 2020).

**Proposition 2.** *The performance of ARIMA (p,d,q) is compared with that of LSTM. The LSTM looks like a smooth, natural competitor to ARIMA (p,d,q) because it can capture a long-term sequence or pattern within sequential data. We start building an LSTM model, which enumerates the stated function f linking $k_t^{(2)}$ to the time lags, as:*

$$k_t^{(2)} = f(k_{t-1}^{(2)}, k_{t-2}^{(2)}, k_{t-3}^{(2)}, k_{t-4}^{(2)}, \dots k_{t-j}^{(2)}) + w_{t-x}^{(3)} \tag{15}$$

*where $j \epsilon \mathbb{N}$ is defined as the number of time lags being considered and $w_{t-x}^{(3)}$ is the homoschedastic error or randomness term.*

**Proof.** The LSTM network, just like many other standard machine learning methods, needs the dataset dividing into testing and training sets. The training set often represents supervised learning, whereas testing is for the validation of the model. Table 1 shows a supervised learning dataset, which is helpful for prediction. Upon completion of training, the network will have learned the input-output functional relationship, thus predicting future values of $k_t^{(2)}$ by using only the input. To be more practical, taking the input as $(m \times J)$ matrix with time lags of $k_t^{(2)}$ as well as the output as the $(m \times 1)$ vector of best current values, with $m \epsilon \mathbb{N}$ is the number as in Table 1. □

**Table 1.** Supervised Learning DataSet.

| | Output | | Input | |
|---|---|---|---|---|
| $k_t^{(2)}$ | $k_{t-1}^{(2)}$ | $k_{t-2}^{(2)}$ | .... | $k_{t-j}^{(2)}$ |
| $k_{t+1}^{(2)}$ | $k_t^{(2)}$ | $k_{t-1}^{(2)}$ | .... | $k_{t-j+1}^{(2)}$ |
| $k_{t+2}^{(2)}$ | $k_{t+1}^{(2)}$ | $k_t^{(2)}$ | .... | $k_{t+j-2}^{(2)}$ |
| $k_{t+3}^{(2)}$ | $k_{t+2}^{(2)}$ | $k_{t+1}^{(2)}$ | .... | $k_{t+j-3}^{(2)}$ |
| .... | .... | .... | .... | .... |
| $k_{t+m}^{(2)}$ | $k_{t+m-1}^{(2)}$ | $k_{t+m-2}^{(2)}$ | .... | $k_{t+m-j}^{(2)}$ |

The predicted $k_t^{(2)}$ values, at time $m + 1, m + 2, m + 3, ..., m + J$, are done recursively. Generally, the predicted values of $k_t^{(2)}$ in a generic time $m + t$ is determined using the values of $k_t^{(2)}$ with $t = (m + \lambda - 1, m + \lambda - 2, m + \lambda - 3, ..., m + \lambda - J)$ as input. The values of $k_t^{(2)}$ are determined by the predicted as opposed to observed values. We start by estimating the CBD model parameters $k_t^{(1)}$, $(x - \bar{x})$ and $k_t^{(2)}$ using the SVD method. The extracted time series of $k_t^{(2)}$ is denoted as the first base for our analysis. The data is then split into training set and testing set as per 80% training and 20% testing rule. Consequently, we determine the last year $T$ of observation. We have done the analysis for the U.K. and Kenya differentiating through gender with one-time lag ($j = 1$) in Table 2.

**Table 2.** Testing set years as per Nations.

| Nations | Number of Years | Years of Testing Set |
|---|---|---|
| U.K. | 1930–2018 | 1998–2018 |
| Kenya | 2010–2020 | 2010–2020 |

When selecting the optimum hyperparameters combination for the neural network, it is essential to carry a preliminary fine-tuning round for all these countries while distinguishing them by gender (see Table 3). In this step, we can get combinations, which will be used during LSTM calibration during the forecasting procedure. On the tuning results, we have discovered that this architecture having one hidden layer does perfume better than others on our data and the number of neurons depending on the country. Using a Rectified Linear Unit (ReLU) as an activation function outperformed many other functions when testing many other countries. Moreover, there is no clear evidence on the influence of the performance of hyper-parameters.

**Table 3.** ARIMA by Nation and Gender.

| Nation | ARIMA Model $(p, d, q)$ |
|---|---|
| U.K. | |
| Males | *ARIMA* (1,1,0) |
| Females | *ARIMA* (1,1,0) |
| Kenya | |
| Males | *ARIMA* (0,1,3) |
| Females | *ARIMA* (0,1,3) |

After the calibration step, the paper's analysis will include numerical and graphical processing and presentation of the goodness of fit. To be specific, the study will follow the approach of out of sample, which denotes the testing step within the field of machine learning. The estimation of parameter $k_t^{(2)}$ parameter is determined using SVD, as for male and female respectively. Figure 3 dashed vertical line shows a separation of the forecasted period compared to one used in training the LSTM network. As for ARIMA models, it

is shown that the confidence interval within 0.995 level of confidence. In addition to the graphical check, we can compare the LSTM performance against those of optimum ARIMA in the testing set before measuring the correctness of the forecasting by calculating the following measures of statistical goodness of fit; which includes Mean Absolute Error (MAE) as well as Root Mean Square Error (RMSE):

$$MAE = \frac{\sum_{\lambda=T+1}^{m-T} |k_\lambda^{(2)} - k_\lambda^{(\hat{2})}|}{(m-T)} \tag{16}$$

$$RMSE = \sqrt{\frac{\sum_{\lambda=T+1}^{m-T} (k_\lambda^{(2)} - k_\lambda^{(\hat{2})})^2}{(m-T)}} \tag{17}$$

Table 4 illustrates the respective performance of ARIMA and LSTM in terms of their RMSE and MAE by the individual nation and gender. From the results of measures of goodness of fit and kt plots, we can see that the LSTM network offer excellent performances when equated to the traditional ARIMA models.

By analyzing error estimates of MAE and RMSE, Kenya shows the best performance LSTM concerning the ARIMA model for both tabulated genders. Moreover, by graphical analysis, the LSTM appears to capture the non-linearity, especially of the future mortality trends, by showing its good capability of bettering representation by decreasing mortality dynamics when dealing with the ARIMA model.

Analytically, we have noticed a higher ability of the LSTM when capturing trends of nonlinear without going in an opposite situation, which is an excessive oscillating or parabolic trend (as well as the latter observed when compared to traditional ANNs). Contrary, the analysis is showing that ARIMA $(p, d, q)$ method is not effective. This evolution of $k_t^{(2)}$ as per ARIMA models is sometimes experienced out of reach within the confidence interval levels, as in the U.K. case for both sexes.

The results obtained have highlighted the ARIMA's inadequacy to detect the ever-decreasing mortality dynamics over time. Though many researchers across the globe still use the ARIMA process when modeling time mortality indexes, because having a fixed structure at the same time works well as long as data satisfies the assumptions of ARIMA like constant variance assumption, which has vast importance for integrated models, it has many flaws when compared to currently existing deep learning techniques. In many cases, life table data may exhibit unpredictable volatility changes for long time series, which doesn't fit the ARIMA assumption well.

Even though ANN is an excellent and outstanding learning algorithm for modeling, it offers the only point of predictions without indicating any form of their variability. In addition, the prediction of confidence intervals is a real substantial challenge within the ANN field. Nevertheless, the LSTM network still demonstrates an excellent candidate to use when predicting the mortality trend accurately over a long time. Table 4 shows that LSTM indeed over-performs the traditional ARIMA (p,d,q) model in all stated nations because of its fantastic architecture, which permits learning the vital influence from historical mortality data and replacing it with high accuracy the future years. The LSTM network capability is seen easily, particularly for the populations of the two nations where $k_t^{(2)}$ parameter does need to take a protuberant linear trend.

**Table 4.** LSTM & ARIMA Performances.

| Nation | | Males | | Females |
|---|---|---|---|---|
| U.K. | MAE | RMSE | MAE | RMSE |
| $k_t^{(2)}$ LSTM | 2.45 | 3.54 | 2.34 | 3.65 |
| $k_t^{(2)}$ ARIMA | 16.56 | 22.45 | 18.45 | 24.45 |
| Kenya | MAE | RMSE | MAE | RMSE |
| $k_t^{(2)}$ LSTM | 2.56 | 3.85 | 2.85 | 3.96 |
| $k_t^{(2)}$ ARIMA | 20.85 | 24.05 | 21.56 | 26.86 |

One remarkable LSTM aspect concerns the probability of achieving optimal predicting performance without resorting to a prior selection, especially of the time steps. For example, we have shown that the determined values of mortality (Figure 3) from logit-mortality rates, $logit\mu_{(x,t)}$ for the Kenyan for males. The ARIMA model offers a trivial forecasted trend shape when compared to LSTM. The straight line of the future $k_t^{(2)}$ values, which varies over time, produces a fundamental behavior of the predicted shape of mortality. On the contrary, the CBD integrated model with the LSTM has an insignificant gap between the real and the forecasted values mortality rates. From the smoothness of the curve, it is easy to prove the capability of LSTM as a better forecast on accurate and big data compared to the historical ARIMA (p,d,q) model.

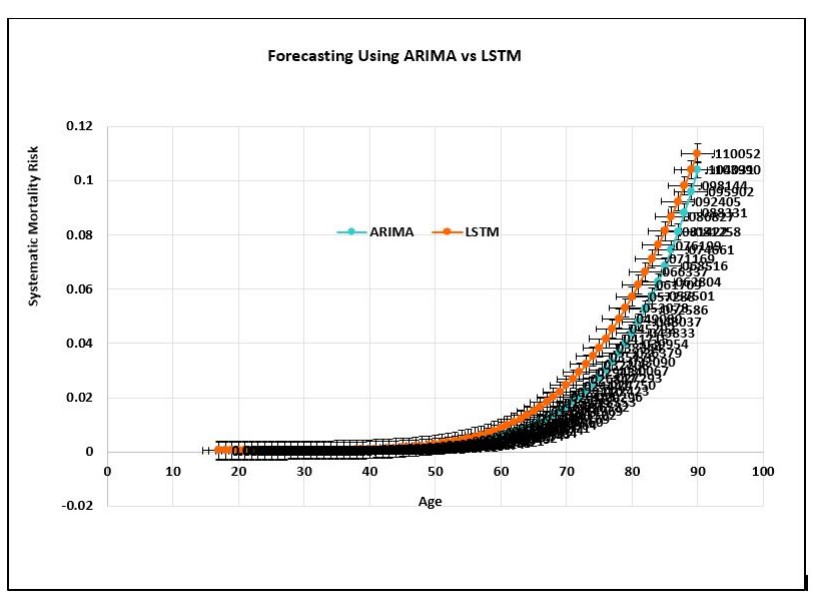

**Figure 3.** Systematic Mortality Risk Prediction Under ARIMA vs. LSTM.

## 5. Conclusions

The values of MAE and RMSE from Table 4, under the LSTM approach, are much lower when compared to those under the traditional ARIMA models for the UK and Kenya. The accuracy of the model when forecasting the mortality models is much better in terms of precision compared to those done on the traditional ARIMA models. Deep learning integrated systematic mortality risk modeling improves the accuracy of the models by approximately 150 % for the two countries.

In this study, we have done a deep learning CBD integrated model based on an RNN possessing LSTM architecture used to predict the future index values of the parameter, $k_t^{(2)}$. This approach has shown very high precision levels when modeling mortality trends and forecasted values as opposed to canonical ARIMA see in Figure 3. In addition, in Figure 3, LSTM has excellent features that offer more accurate forecasting while decreasing

mortality trends over time than the best conventional ARIMA (p,d,q) process. In addition, its powerful effect is replicated to the trends of complete expectations of life expectancy and death distribution during a person's lifetime.

This LSTM has a highly accurate non-linearity behavior, making it easy to determine the differences between predicted and actual values over time. The model offers a more optimistic scenario when modeling systematic mortality risk.

## 6. Recommendations

Using a CBD model, we have demonstrated that deep learning techniques make the model more accurate when modeling and predicting, thus reducing the challenges associated with errors. Any government or private company that needs to model a behaviour study can apply deep learning as opposed to traditional statistical estimations.

However, training on the use of machine learning needs to be done for the professionals in these institutions since this will enable them to save huge lots of money when pricing the financial products based on a prediction and projections. For instance, the valuation of life assurance products such as assurances and annuities depends on predicted systematic mortality risk levels, meaning that poor estimation of the risk can lead to the insurance, pension, and social security firms making substantial financial losses.

On implementing the LSTM method in a policy document by the Kenyan government, the information should be fed into the input layer before transferred to its hidden layer. The interconnections of the Neural Networks between the two layers assign proportions or weights to every input randomly.

**Author Contributions:** Conceptualization, J.O., P.N. and P.W.; Formal analysis, J.O.; Investigation, P.W.; Methodology, J.O.; Project administration, P.N.; Resources, P.W.; Supervision, P.W.; Validation, P.N.; Writing—original draft, J.O.; Writing—review and editing, P.N. and P.W. All authors have read and agreed to the published version of the manuscript.

**Funding:** This research had no external funding.

**Informed Consent Statement:** Not applicable.

**Conflicts of Interest:** All authors of this have declared that they have no hidden conflicts of interest.

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
