# Peer review of "A Deep Learning Integrated Cairns-Blake-Dowd (CBD) Sytematic Mortality Risk Model"

_jrfm, doi:10.3390/jrfm14060259_

Round 1

Reviewer 1 Report

This article is about "stochastic mortality models". To predict the trends of mortality series over a period of time of the model, Authorsl appled the Recurrent Neural Networks within a Long Short-Term Memory or just an artificial LSTM architecture.

The article is interesting, but it has some disadvantages.
In summary, no "research purpose" and "own contribution".
In its present form, the purpose of the study and own contribution are illegible.
Conclusions from the study should be in the form of graphs and comparative tables.
Mathematical formulas should be written correctly. The article should be improved according to the publisher's requirements. Math formulas are very poorly written.

Author Response

Thank you for your comments.

  1. The document as originally written in Latex, before converting it a word document.  This has distorted the document making it unreadable. I believe the original latex Document is perfectly readable.
  2. The research purpose is model mortality risk using the deep learning technique which offers a better prediction accuracy when compared to the traditional prediction methods. My purpose was to illustrate that deep learning is superior in prediction.
  3. The tables shows the illustration how the method works well when compared to the classical models of prediction.

Reviewer 2 Report

In this paper the authors introduce the LSTM and RNN architectures within the normal scheme of the CBD model for predicting the future index values of on the time index kt(2) trend, that is the slope of the model CBD in year t. They argue that the LSTM network is an excellent candidate to use when predicting accurately the trend of mortality over a long time. They also offer some recommendation to be followed by analysts and policy makers, and pension schemes.

In general, the idea of the paper is interesting and contributes to the relevant literature regarding mortality rate forecasting.

1). However, I believe that the paper should be re-written. First the mathematical presentation is not acceptable. I do not understand most of the symbols. For instance, in the presentation of the CBD model, or in ANN (example equation 0.9 among others) or for the root mean square error (0.17) … and many others throughout the whole paper. If a reader is not familiar with the particular models and statistical techniques, it would impossible to understand.

2). In fact, the study of mortality risk is crucial for both public agencies and private actuaries. Cairns et al. (2009) and in https://www.macs.hw.ac.uk/~andrewc/papers/ajgc53.pdf provide a detailed description of the most important stochastic mortality models, and Lee–Carter (LC) and the Cairns–Blake–Dowd (CBD) are two of the major. The choice of the CBD model should be discussed and explained in the paper. See Carlo Maccheroni, and Samuel (2017). Backtesting the Lee–Carter and the Cairns–Blake–Dowd Stochastic Mortality Models on Italian Death Rates, Risks, 5 (34). doi:10.3390/risks5030034 www.mdpi.com/journal/risks

3). Isn’t too much complicated to be followed by practitioners? I suggest the authors to present in simple terms the steps to be followed. This could be done in the section 6. Recommendations.

Author Response

Thank you for your comments.

  1. The mathematical equations were distorted before they were written in word document as opposed to PDF. My PDF document with the mathematical formulas were good, which I have shared written in Latex.
  2. I have reviewed the papers recommended by the reviewer and they have helped me improve the document.
  3. I have provided a step to step plan how to implement the idea for the policymakers such as the Kenyan government or pension scheme firms.

Reviewer 3 Report

Dear Authors,

the manuscript is very hard to decipher. I would strongly recommend checking your submission before uploading.

The abstract contains no structure. More then half of the abstract is description of other papers. Maybe a slight structure of background, objectives, methods, results would help to convey your research in a more adequate way.

Skimming the article led me to the conclusion that the formulas and equations are not readable, even paragraph signs are present. Notation therefore confusing for the reader.

The sole figure present in the manuscript is haphazard and lines can not be distinguished from each other.

Author Response

Thank you for your comments.

  1. The document as originally written in Latex, before converting it a word document.  This has distorted the document making it unreadable. I believe the original latex Document is perfectly readable.
  2. The research objective is model mortality risk using the deep learning technique which offers a better prediction accuracy when compared to the traditional prediction methods. The results were illustrated and discussed in chapter 4.
  3. The tables show the illustration how the method works well when compared to the classical models of prediction.

Round 2

Reviewer 2 Report

The revised form of the paper is satisfactory.

Thank you.